# Shedding Light on Large Generative Networks:
# Estimating Epistemic Uncertainty in Diffusion Models

**Lucas Berry**[1,2]                **Axel Brando**[3]                **David Meger**[1,2]

[1]School of Computer Science, McGill University, Montreal, Quebec, Canada
[2]Centre for Intelligent Machines, McGill University, Montreal, Quebec, Canada
[3]Barcelona Supercomputing Center - Centro Nacional de Supercomputación (BSC-CNS), Spain

## Abstract

Generative diffusion models, notable for their large parameter count (exceeding 100 million) and operation within high-dimensional image spaces, pose significant challenges for traditional uncertainty estimation methods due to computational demands. In this work, we introduce an innovative framework, Diffusion Ensembles for Capturing Uncertainty (DECU), designed for estimating epistemic uncertainty for diffusion models. The DECU framework introduces a novel method that efficiently trains ensembles of conditional diffusion models by incorporating a static set of pretrained parameters, drastically reducing the computational burden and the number of parameters that require training. Additionally, DECU employs Pairwise-Distance Estimators (PaiDEs) to accurately measure epistemic uncertainty by evaluating the mutual information between model outputs and weights in high-dimensional spaces. The effectiveness of this framework is demonstrated through experiments on the ImageNet dataset, highlighting its capability to capture epistemic uncertainty, specifically in under-sampled image classes.

## 1 INTRODUCTION

In this paper, we introduce Diffusion Ensembles for Capturing Uncertainty (DECU), a novel approach designed to quantify epistemic uncertainty in conditioned diffusion models that generate high-dimensional images ($256 \times 256 \times 3$). To the best of our knowledge, our method is the first in addressing the challenge of capturing epistemic uncertainty in conditional diffusion models for image generation. Figure 1 illustrates an example of DECU generating images. In sub-figure (a), a class label with low epistemic uncertainty results in images closely resembling their class, while in

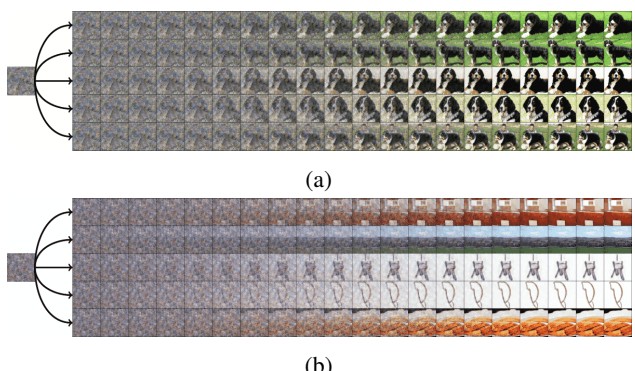

(a)

(b)

Figure 1: Image generation progression through DECU, each row refers to an ensemble component, for the class label of Bernese mountain dog with low epistemic uncertainty (a) and moving van with high epistemic uncertainty (b).

sub-figure (b), a class label with high epistemic uncertainty leads to images that do not resemble their respective class.

DECU employs two key strategies. Firstly, it efficiently trains an ensemble of diffusion models within a subset of the network. This is achieved through the utilization of pretrained networks from Rombach et al. [2022]. Training an ensemble of diffusion models in a naive manner would demand substantial computational resources, considering that each model encompasses hundreds of millions of parameters and requires weeks to train [Dhariwal and Nichol, 2021].

Secondly, DECU incorporates Pairwise-Distance Estimators (PaiDEs), a non-sample-based method proven effective in estimating the mutual information between the model's output and its weights in high-dimensional regression tasks [Kolchinsky and Tracey, 2017, Berry and Meger, 2023a]. The mutual information between model weights and output is a well-established metric for measuring epistemic uncertainty [Houlsby et al., 2011]. PaiDEs capture this mutual information by assessing consensus among ensemble

components through the distributional distance between each pair of components. Distributional distance serves as a metric to gauge the similarity between two probability distributions.

Epistemic uncertainty stems from a model's ignorance and can be reduced with more data, while aleatoric uncertainty arises from inherent randomness in the data (e.g. when some crucial variables are hidden) and is thus irreducible [Hora, 1996, Der Kiureghian and Ditlevsen, 2009, Hüllermeier and Waegeman, 2021]. With the increasing integration of large diffusion models into automated systems [Rombach et al., 2022, Dhariwal and Nichol, 2021], gaining a comprehensive understanding of the images generated by these black-box models becomes paramount. Generative image models play a crucial role in diverse applications, notably in medical image generation and self-driving systems [Guibas et al., 2017, Kazerouni et al., 2022, Hu et al., 2023]. Both of these domains are riddled with uncertainty, capable of yielding catastrophic outcomes for human life. Our approach illuminates the black box of diffusion models by estimating their epistemic uncertainty, offering assistance in situations where predictions from automated systems are more uncertain. In addition, our proposed framework can be used to build solutions that satisfy international safety standards for automated systems (self-driving ISO/IEC 26262:2011 Salay et al. [2018] or the generic AI systems ISO/IEC 23053:2022).

By combining our efficient ensemble technique for diffusion models with PaiDEs, we address the challenge of capturing epistemic uncertainty in conditional diffusion models for image generation. We evaluate DECU on the ImageNet dataset Russakovsky et al. [2015], and our contributions can be summarized as follows:

- We establish the framework of DECU for class-conditioned diffusion models (Section 3).
- We assess the effectiveness of DECU on image generation on the ImageNet dataset, a commonly used but significantly challenging benchmark within the community (Section 4.1).
- We provide an evaluation of image diversity within DECU (Section 4.2).

These advancements illuminate the previously opaque area of epistemic uncertainty in conditional diffusion models, offering significant implications for decision-making processes and risk evaluation.

## 2 BACKGROUND

Diffusion models create a Markov chain, where, at each transition, they sample from a Gaussian distribution. This inherent feature makes them particularly suitable for uncertainty estimation, as the Gaussian probability distributions provide a natural framework for reasoning about uncertainty [Hüller-

meier and Waegeman, 2021]. PaiDEs present an efficient method for estimating epistemic uncertainty by utilizing established pairwise distance formulas between Gaussian components within the ensemble.

### 2.1 PROBLEM STATEMENT & DIFFUSION MODELS

In the context of supervised learning, we define a dataset $\mathcal{D} = \{x_i, y_{i,0}\}_{i=1}^{N}$, where $x_i$ represents class labels, and each $y_{i,0}$ corresponds to an image with dimensions of $256 \times 256 \times 3$. Our primary goal is to estimate the conditional probability $p(y|x)$, which is complex, high-dimensional, continuous, and multi-modal.

To effectively model $p(y|x)$, we utilize diffusion models, which have gained significant recognition for their ability to generate high-quality images [Rombach et al., 2022, Saharia et al., 2022]. These models employ a two-step approach referred to as the forward and reverse processes to generate realistic images. Please note that we will omit the subscript $i$ from $y_{i,0}$ and $x_i$ for simplicity in notation. In the forward process, an initial image $y_0$ undergoes gradual corruption through the addition of Gaussian noise in $T$ steps, resulting in a sequence of noisy samples $y_1, y_2, \ldots, y_T$:

$$q(y_t|y_{t-1}) = \mathcal{N}(y_t; \sqrt{1 - \beta_t}y_{t-1}, \beta_t\mathbf{I})$$
$$q(y_{1:T}|y_0) = \prod_{t=1}^{T} q(y_t|y_{t-1}),$$

where $\beta_t \in (0, 1)$ and $\beta_1 < \beta_2 < \ldots < \beta_T$. The forward process draws inspiration from non-equilibrium statistical physics [Sohl-Dickstein et al., 2015].

The reverse process aims to remove noise from the corrupted images and reconstruct the original image, conditioned on the class label. This is accomplished by estimating the conditional distribution $q(y_{t-1}|y_t, x)$ using the model $p_\theta$. The reverse diffusion process can be represented as follows:

$$p_\theta(y_{0:T}|x) = p(y_T) \prod_{t=1}^{T} p_\theta(y_{t-1}|y_t, x) \tag{1}$$
$$p_\theta(y_{t-1}|y_t, x) = \mathcal{N}(y_{t-1}; \mu_\theta(y_t, t, x), \Sigma_\theta(y_t, t, x)).$$

In this formulation, $p_\theta(y_{t-1}|y_t, x)$ represents the denoising distribution parameterized by $\theta$, which follows a Gaussian distribution with mean $\mu_\theta(y_t, t, x)$ and covariance matrix $\Sigma_\theta(y_t, t, x)$. Note that $\mu_\theta(y_t, t, x)$ and $\Sigma_\theta(y_t, t, x)$ are learned models. The forward and reverse diffusion processes each create a Markov chain to generate images.

To model the reverse process $p_\theta$, calculating the exact log-likelihood $\log(p_\theta(y_0|x))$ is typically infeasible. This necessitates the use of the evidence lower bound (ELBO), a technique reminiscent of variational autoencoders (VAEs) [Kingma and Welling, 2013]. The ELBO can be expressed

as follows:

$$-\log(p_\theta(y_0|x)) \leq -\log(p_\theta(y_0|x)) \\ + D_{KL}(q(y_{1:T}|y_0) \parallel p_\theta(y_{1:T}|y_0, x)). \quad (2)$$

The loss function in Equation 2 represents the trade-off between maximizing the log-likelihood of the initial image and minimizing the KL divergence between the true posterior $q(y_{1:T}|y_0)$ and the approximate posterior $p_\theta(y_{1:T}|y_0, x)$. Equation 2 can be simplified using the properties of diffusion models. For a more comprehensive introduction to diffusion models, please refer to Ho et al. [2020].

## 2.2 EPISTEMIC UNCERTAINTY AND PAIDES

Probability theory provides a natural framework to reason about uncertainty [Cover and Thomas, 2006, Hüllermeier and Waegeman, 2021]. In the context of capturing uncertainty from a conditional distribution, a widely used metric is that of conditional differential entropy,

$$H(y_{t-1}|y_t, x) = -\int p(y_{t-1}|y_t, x) \ln p(y_{t-1}|y_t, x) dy_t.$$

Leveraging conditional differential entropy Houlsby et al. [2011] defined epistemic uncertainty as follows,

$$I(y_{t-1}, \theta|y_t, x) = H(y_{t-1}|y_t, x) \\ - E_{p(\theta)}\left[H(y_{t-1}|y_t, x, \theta)\right], \quad (3)$$

where $I(\cdot)$ denotes mutual information and $\theta \sim p(\theta)$. Mutual information measures the information gained about one variable by observing the other. When all of $\theta$'s produce the same $p_\theta(y_0|y_T, x)$, $I(y_{t-1}, \theta|y_t, x)$ is zero, indicating no epistemic uncertainty and that each component agrees about the output distribution. Conversely, when said distributions have non-overlapping supports, epistemic uncertainty is high and each ensemble component disagrees strongly about the output distribution.

A distribution over weights becomes essential for estimating $I(y_{t-1}, \theta|y_t, x)$. One effective approach for doing this is through the use of ensembles. Ensembles harness the collective power of multiple models to estimate the conditional probability by assigning weights to the output from each ensemble component. This can be expressed as follows:

$$p_\theta(y_{t-1}|y_t, x) = \sum_{j=1}^{M} \pi_j p_{\theta_j}(y_{t-1}|y_t, x) \quad \sum_{j=1}^{M} \pi_j = 1, \quad (4)$$

where $M$, $\pi_j$ and $\theta_j$ denote the number of model components, the component weights and different component parameters, respectively. Note that the model components are assumed to be uniform, $\pi_j = \frac{1}{M}$, as this approach has been demonstrated to be effective for estimating epistemic uncertainty [Chua et al., 2018, Berry and Meger, 2023a]. When creating an ensemble, two common approaches are typically

considered: randomization [Breiman, 2001] and boosting [Freund and Schapire, 1997]. While boosting has paved the way for widely adopted machine learning methods [Chen and Guestrin, 2016], randomization stands as the preferred choice in the realm of deep learning due to its tractability and straightforward implementation [Lakshminarayanan et al., 2017].

In the context of continuous outputs and ensemble models, Equation 3 often does not have a closed-form solution due to the left hand-side:

$$H(y_{t-1}|y_t, x) = \int \sum_{j=1}^{M} \pi_j p_{\theta_j}(y_{t-1}|y_t, x) \\ \times \ln \sum_{j=1}^{M} \pi_j p_{\theta_j}(y_{t-1}|y_t, x) dy_t.$$

Thus, previous methods have relied on Monte Carlo (MC) estimators to estimate epistemic uncertainty [Depeweg et al., 2018, Postels et al., 2020]. MC estimators are convenient for estimating quantities through random sampling and are more suitable for high-dimensional integrals compared to other numerical methods. However, as the number of dimensions increases, MC methods typically require a larger number of samples [Rubinstein and Glynn, 2009].

Given that our output is very high-dimensional, MC methods become extremely computationally demanding, necessitating an alternative approach. For this, we rely on Pairwise-Distance Estimators (PaiDEs) to estimate epistemic uncertainty [Kolchinsky and Tracey, 2017]. PaiDEs have been shown to accurately capture epistemic uncertainty for high-dimensional continuous outputs [Berry and Meger, 2023a]. Let $D(p_i \parallel p_j)$ denote a generalized distance function between the probability distributions $p_i$ and $p_j$, where $p_i$ and $p_j$ represent $p_i = p(y_{t-1}|y_t, x, \theta_i)$ and $p_j = p(y_{t-1}|y_t, x, \theta_j)$, respectively. More specifically, $D$ is referred to as a premetric, satisfying $D(p_i \parallel p_j) \geq 0$ and $D(p_i \parallel p_j) = 0$ if $p_i = p_j$. The distance function need not be symmetric nor obey the triangle inequality. As such, PaiDEs can be defined as follows:

$$\hat{I}_\rho(y_{t-1}, \theta|y_t, x) = -\sum_{i=1}^{M} \pi_i \ln \sum_{j=1}^{M} \pi_j \exp\left(-D(p_i \parallel p_j)\right)$$

PaiDEs offer a variety of options for $D(p_i \parallel p_j)$, such as Kullback-Leibler divergence, Wasserstein distance, Bhattacharyya distance, Chernoff $\alpha$-divergence, Hellinger distance and more.

## 3 METHODOLOGY

Diffusion models come with a substantial training cost, requiring 35 V100 days for latent diffusion class-conditioned models on ImageNet [Rombach et al., 2022]. Naively training $M$ distinct diffusion models only worsens this computational load. To address this challenge, we propose training

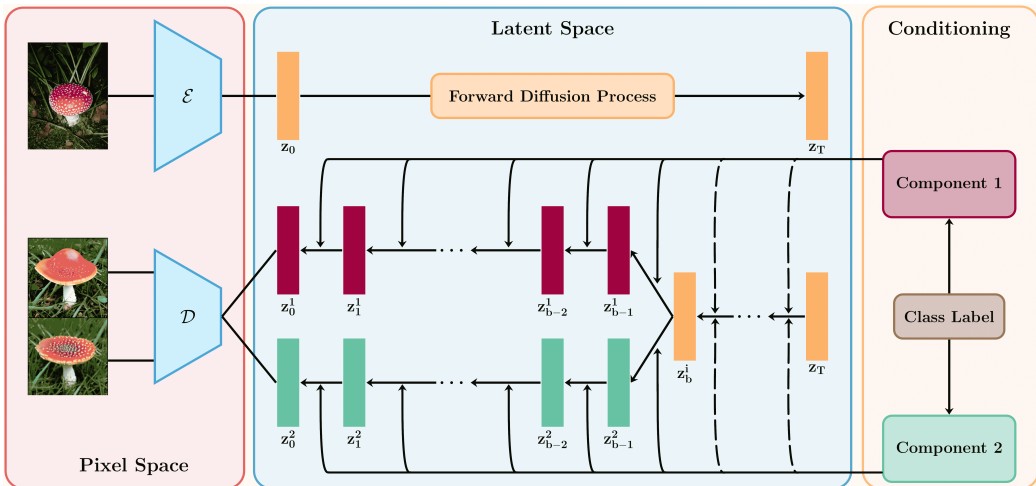

Figure 2: The ensemble pipeline for DECU, shown here with two components. During the reverse process, the previous latent vector $z_t^j$ passes through a UNet to yield $z_{t-1}^j$. Dashed lines signify the random selection of one ensemble component for rollout until the branching point. Our ensembles are constructed within the embedding layer, which accepts the class label as input. We create diversity through random initialization and by training each component on different subsets of the data. The encoders, decoders, and UNets for each component are shared, and we leverage pretrained networks from Rombach et al. [2022]. Notably, this reduces the number of parameters required for training from 456 million to 512 thousand.

a sub-module within the diffusion model architecture and show that this is adequate for estimating epistemic uncertainty. Furthermore, there are multiple junctures within the reverse diffusion process where one could effectively estimate uncertainty. We demonstrate the specific point at which this estimator yields accurate estimates.

## 3.1 DIFFUSION ENSEMBLES

We employ the latent diffusion models introduced by Rombach et al. [2022] to construct our ensembles. They proposed the use of an autoencoder to learn the diffusion process in a latent space, significantly reducing sampling and training time compared to previous methods by operating in a lower-dimensional space, $z_t$, which is $64 \times 64 \times 3$. Using this framework we can estimate epistemic uncertainty in this lower-dimensional space,

$$
\hat{I}_\rho(z_{t-1}, \theta | z_t, x) = -\sum_{i=1}^{M} \pi_i
$$
$$
\times \ln \sum_{j=1}^{M} \pi_j \exp\left(-D(p_i \parallel p_j)\right),
$$

(5)

where $p_i$ and $p_j$ now denote Gaussians in the latent space. This approach is akin to previous methods that utilize latent spaces to facilitate the estimation of epistemic uncertainty [Berry and Meger, 2023b].

To fit our ensembles, we make use of pre-trained weights for the UNet and autoencoder from Rombach et al. [2022],

keeping them static throughout training. The only part of the network that is trained is the conditional label embedding layer, which is randomly initialized for each ensemble component. This significantly reduces the number of parameters that need to be trained (512k instead of 456M) as well as the training time (by 87%), compared to training a full latent diffusion model on ImageNet. It is important to note that each ensemble component can be trained in parallel, as the shared weights remain static for each component, further enhancing training efficiency.

Upon completion of the training process, we utilize the following image generation procedure:

1. Sample random noise $z_T$ and an ensemble component $p_j$.

2. Use $p_j$ to traverse the Markov chain until reaching step $b$, our branching point.

3. Branch off into $M$ separate Markov chains, each associated with a different component.

4. Progress through each Markov chain until reaching step 0, $z_0^j$, and then decoding each $z_0^j$ to get $y_0^j$.

Figure 2 illustrates the described pipeline with two components. Note that during the reverse process the previous latent vector $z_t^j$, the time step $t$ and the output from component $j$ are passed through a UNet to arrive at $z_{t-1}^j$. By leveraging the inherent Markov chain structure within the diffusion model, we can examine image diversity at different branching points. Note that our loss function for training

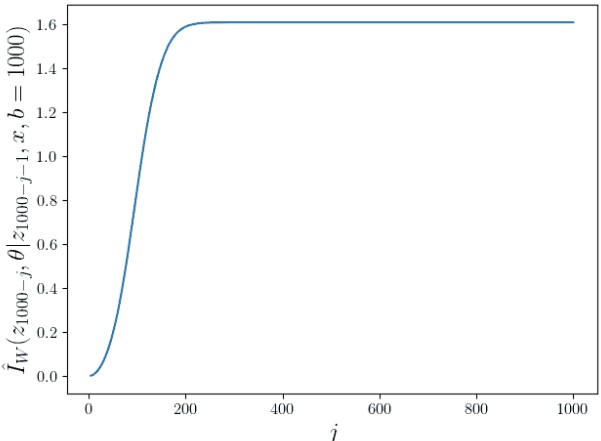

Figure 3: Our estimator for epistemic uncertainty increases with distance from the branch point, converging to $-\ln\frac{1}{5} \approx 1.609$.

each component is the same as Rombach et al. [2022]. We utilize an ensemble of 5 components, a number we found to be sufficient for estimating epistemic uncertainty. For additional hyperparameter details, refer to Appendix A.

### 3.2 DIFFUSION ENSEMBLES FOR UNCERTAINTY

Diffusion models yield a Gaussian distribution at each step during the reverse process, as shown in Equation 1. One can estimate epistemic uncertainty at any $t$ beyond the branching point $b$; however, the further away from the branching point epistemic uncertainty is estimated, the more the Gaussian distributions diverge from one another. Consequently, when PaiDEs are applied in this scenario, they will converge to $-\ln\frac{1}{M}$. This behavior occurs because as the Gaussians diverge more and more, the distance measure, $D(p_i \parallel p_j)$, grows which implies $\exp(-D(p_i \parallel p_j))$ tends to 0. Figure 3 shows this relationship in our context. Therefore, to estimate $I(z_{t-1}, \theta|z_t, x, b = t)$, we utilize PaiDEs right after the branching point as we found this sufficient to estimate epistemic uncertainty.

To generate images, we utilize denoising diffusion implicit models (DDIM) with 200 steps, following the training of a diffusion process with $T = 1000$. DDIM enables more efficient image generation by permitting larger steps in the reverse process without altering the training methodology for diffusion models [Song et al., 2020]. Furthermore, in the DDIM implementation by Rombach et al. [2022], the covariance, $\Sigma_\theta(z_t, t, x)$, is intentionally set to a zero matrix, irrespective of its inputs, aligning with the approach in Song et al. [2020]. However, this prevents us from using

KL-Divergence and Bhattacharyya distance, which are undefined in this case. Therefore, we propose a novel PaiDE using the 2-Wasserstein Distance, which is well-defined between Gaussians in such cases. This distance can be expressed as:

$$W_2(p_i \parallel p_j) = ||\mu_i - \mu_j||_2^2 \\ + \text{tr}\left[\Sigma_i + \Sigma_j - 2\left(\Sigma_i^{1/2}\Sigma_j\Sigma_i^{1/2}\right)^{1/2}\right], \quad (6)$$

where $p_i \sim N(\mu_i, \Sigma_i)$ and $p_j \sim N(\mu_j, \Sigma_j)$. When $\Sigma_i$ and $\Sigma_j$ are zero matrices, it yields the following estimator:

$$\hat{I}_W(z_{t-1}, \theta|z_t, x, b = t) = -\sum_{i=1}^{M} \pi_i \\ \times \ln \sum_{j=1}^{M} \pi_j \exp\left(-W_2(p_i \parallel p_j)\right), \quad (7)$$

$$W_2(p_i \parallel p_j) = ||\mu_i - \mu_j||_2^2.$$

This combination of ensemble creation and epistemic uncertainty estimation encapsulates DECU.

## 4  EXPERIMENTAL RESULTS

The experiments in this study assessed the DECU method by utilizing the ImageNet dataset, a comprehensive collection comprising 1.28 million images distributed across 1000 classes. ImageNet is recognized as a challenging benchmark dataset for large generative models [Brock et al., 2018, Dhariwal and Nichol, 2021]. To evaluate the performance of DECU, a specific subset called the *binned classes* dataset was carefully curated in order to assess epistemic uncertainty estimates. This subset included 300 classes divided into distinct bins: 100 classes for bin 1, another 100 for bin 10, and an additional 100 for bin 100. The remaining 700 classes were grouped into bin 1300. For each ensemble component, a dataset was formed with the following selection process:

- 1 random image per class from bin 1.
- 10 random images per class from bin 10.
- 100 random images per class from bin 100.
- All 1300 images per class from bin 1300 were utilized.

Throughout the training process, each ensemble component was exposed to a total of 28,162,944 images, accounting for repeated images across training epochs. It is worth noting that this stands in contrast to the 213,600,000 images required to train an entire network from scratch for class-conditioned ImageNet models [Rombach et al., 2022].

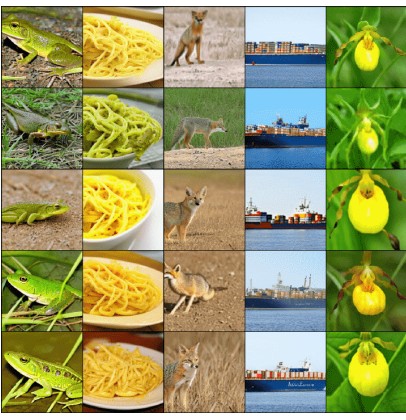 VS 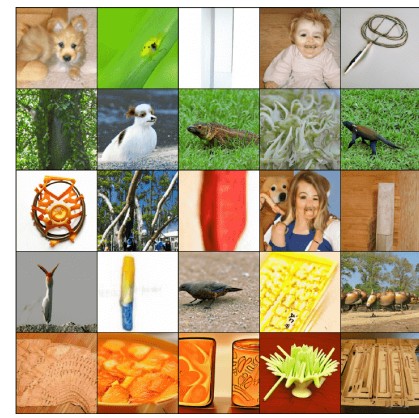

Figure 4: The left image displays low epistemic uncertainty image generation (bin 1300) for five class labels: bullfrog, carbonara, grey fox, container ship, and yellow lady's slipper. The right image shows high epistemic uncertainty image generation (bin 1) for cleaver, Sealyham terrier, lotion, shoji, and whiskey jug. Each row represents an ensemble component with $b = 1000$.

## 4.1 RECOGNITION OF UNDERSAMPLED CLASSES

In this section, we assess the capability of our framework to distinguish classes with limited training images using the *binned classes* dataset. Notably, bins with lower values produced lower-quality images, as illustrated in Figure 4. This figure showcases images with lower epistemic uncertainty generated from five classes in bin 1300 on the left, and images with greater uncertainty generated from five classes in bin 1 on the right. Each row corresponds to an ensemble component, and we set $b = 1000$. The visual contrast highlights a clear trend: with a higher number of training images in bin 1300, our framework produces images that closely align with the respective class labels. This observation is further supported by Figure 9 in the Appendix, which presents another illustrative example of the same trend.

Furthermore, we compute $\hat{I}_W(z_0, \theta | z_5, x, b = 5)$ for each class. To do this, we randomly select 8 samples of random noise and use $b = 5$. It's important to note that we can only take steps of 5 through the diffusion process due to the 200 DDIM steps. We then average the ensemble's epistemic uncertainty over these 8 random noise samples. Figure 5 illustrates the distributions of epistemic uncertainty for each bin. The distributions for the larger bins are skewed more towards 0 compared to the smaller bins. This trend is also reflected in the mean of each distribution, represented by the dashed lines. These findings demonstrate that DECU can effectively measure epistemic uncertainty on average for class-conditioned image generation.

Additionally to estimating the overall uncertainty of a given class, we analyze per-pixel uncertainty in a generated image. We treat each pixel as a separate Gaussian and apply our

| $b$ | 1 | 10 | 100 | 1300 |
|------|-----------|-----------|-----------|-----------|
| 1000 | $0.36 \pm 0.09$ | $0.37 \pm 0.09$ | $0.41 \pm 0.10$ | $\mathbf{0.51} \pm 0.13$ |
| 750 | $0.50 \pm 0.14$ | $0.51 \pm 0.14$ | $0.54 \pm 0.14$ | $\mathbf{0.63} \pm 0.13$ |
| 500 | $0.64 \pm 0.13$ | $0.64 \pm 0.13$ | $0.67 \pm 0.11$ | $\mathbf{0.76} \pm 0.09$ |
| 250 | $0.92 \pm 0.05$ | $0.92 \pm 0.05$ | $0.92 \pm 0.04$ | $\mathbf{0.94} \pm 0.03$ |

Table 1: SSIM calculated between all pairs of generated images per class at different values of $b$ across each bin. Results shown are mean $\pm$ one standard deviation. Higher values indicate greater similarity and the highest mean in each row is bolded.

estimator on a pixel-by-pixel basis. It's worth noting that we first map from the latent vector to image space, so we are estimating epistemic uncertainty in image space and then average across the three channels. An example of this procedure can be seen in Figure 6. For bin 1300, we observe that epistemic uncertainty highlights different birds that could have been generated from our ensemble. Furthermore, bins with lower values exhibit a higher density of yellow, indicating greater uncertainty about what image to generate. Two additional examples contained in the Appendix, Figure 10 and Figure 11, display the same patterns.

## 4.2 IMAGE DIVERSITY BETWEEN COMPONENTS

Apart from assessing image uncertainty, we also conducted an analysis of image diversity across the ensemble with respect to different branching points. To gauge this diversity, we generated images using our framework and computed the Structural Similarity Index Measure (SSIM) between every pair of generated images produced by each compo-

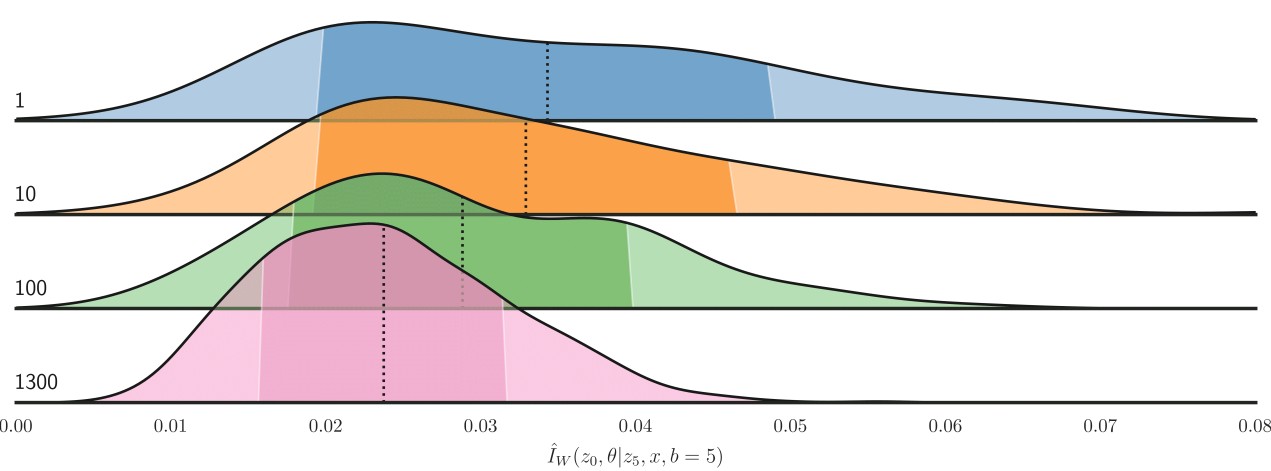

$\hat{I}_W(z_0, \theta | z_5, x, b = 5)$

Figure 5: This figure displays uncertainty distributions for each bin, derived from corresponding class uncertainty estimates.

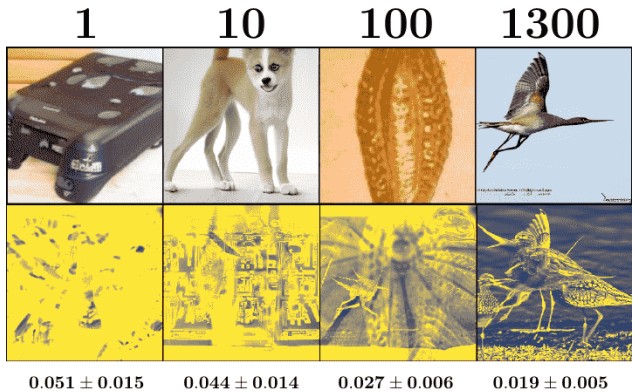

Figure 6: Pixel uncertainty (yellow for high, blue for low) shown for one class in each bin (left to right: wall clock, head cabbage, rubber eraser, Red Shank bird). Numbers below images indicate mean estimated $\hat{I}_W(z_0, \theta | z_5, x, b = 5) \pm$ one standard deviation.

nent. The results can be found in Table 1. Notably, bins with larger values produced images that were more similar. This is attributed to the fact that ensemble components learned to better represent classes in the bin with larger values, resulting in greater agreement amongst the ensemble components. Furthermore, as the branching point increases, the images become more dissimilar. This phenomenon arises because, with a higher $b$, each ensemble component progresses further through the reverse process independently, leading to greater image variation. Visualizations of this phenomenon

can be seen in Figure 7 and Figure 8, where the variety in image generation clearly dissipates as the branching point decreases. Additional visualizations are contained in the Appendix (Figure 12 and Figure 13).

## 5 RELATED WORKS

Constructing ensembles of diffusion models is challenging due to the large number of parameters, often in the range of hundreds of millions [Saharia et al., 2022]. Despite this difficulty, methods such as eDiff-I have emerged, utilizing ensemble techniques to improve image fidelity [Balaji et al., 2022]. In contrast, our approach specifically targets the measurement of epistemic uncertainty.

Previous research has employed Bayesian approximations for neural networks in conjunction with information-based criteria to tackle the problem of epistemic uncertainty estimation in image classification tasks [Gal et al., 2017, Kendall and Gal, 2017, Kirsch et al., 2019]. These works apply epistemic uncertainty estimation to simpler discrete output spaces. In addition to Bayesian approximations, ensembles are another method for estimating epistemic uncertainty [Lakshminarayanan et al., 2017, Choi et al., 2018, Chua et al., 2018]. They have been used to quantify epistemic uncertainty in regression problems [Depeweg et al., 2018, Postels et al., 2020, Berry and Meger, 2023b,a]. Postels et al. [2020] and Berry and Meger [2023b] develop efficient ensemble models based on Normalizing Flows (NF) that accurately capture epistemic uncertainty. Berry and Meger [2023a] advances these findings by utilizing PaiDEs

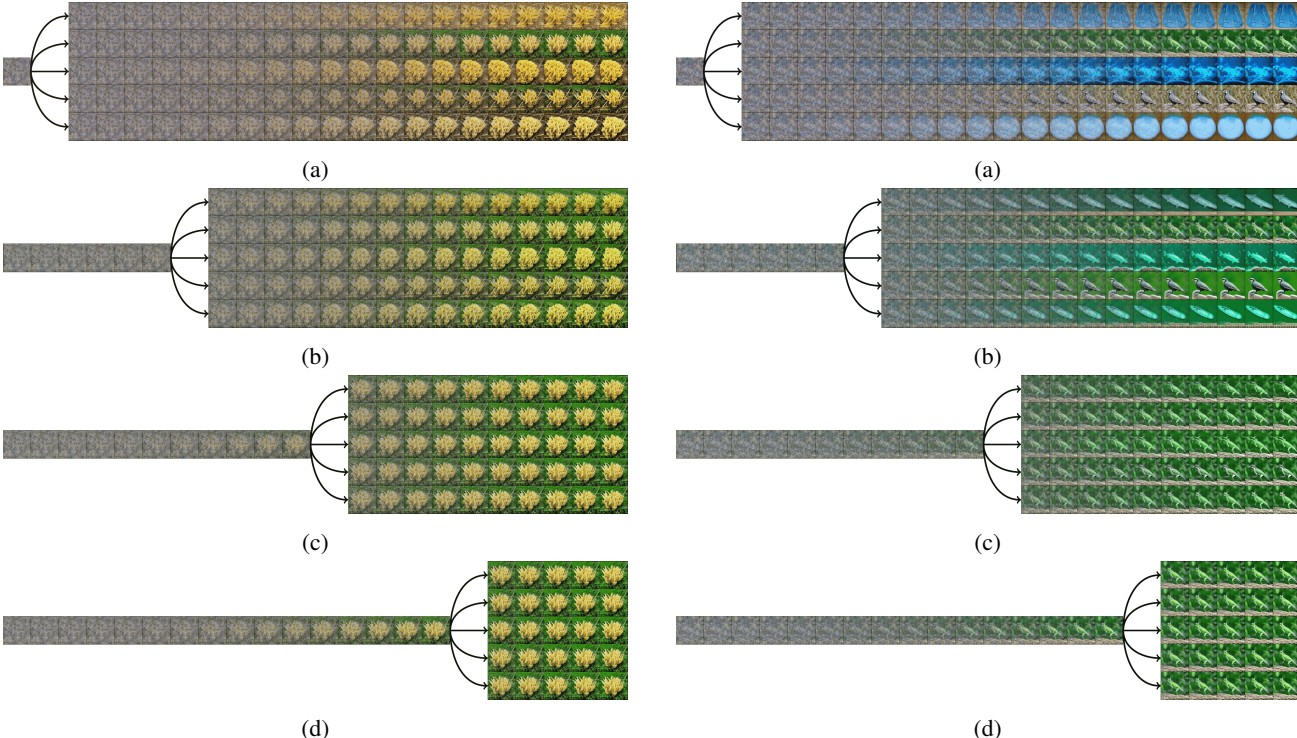

Figure 7: Image generation progression through DECU for the class label coral fungus from bin 1300 for each branching point: (a) 1000, (b) 750, (c) 500, (d) 250.

Figure 8: Image generation progression through DECU for the class label monastery from bin 1 for each branching point: (a) 1000, (b) 750, (c) 500, (d) 250.

to estimate epistemic uncertainty on 257-dimensional output space with normalizing flows. Our work builds on this line of research by showcasing how to extend these methods to higher-dimensional outputs (196,608 dimensions) and for large generative diffusion models.

To capture epistemic uncertainty, we employ the mutual information between model outputs and model weights [Houlsby et al., 2011]. This metric has previously been utilized for data acquisition in active learning settings, notably in BALD [Houlsby et al., 2011] and BatchBALD [Kirsch et al., 2019]. Applying such techniques to diffusion models is well justified, as collecting data for image generation models proves to be a costly endeavor. However, currently, it is infeasible to do active learning for large diffusion models due to the high computational costs associated with training after each acquisition batch. Anticipating future advancements in computational resources holds the promise of increased feasibility to explore these ideas. This underscores another potential use case for epistemic uncertainty in diffusion models.

In addition to PaiDEs, various methods have emerged for estimating epistemic uncertainty without relying on sampling [Van Amersfoort et al., 2020, Charpentier et al., 2020]. Van Amersfoort et al. [2020] and Charpentier et al. [2020]

primarily focus on classification tasks. While Charpentier et al. [2021] extends tackle regression tasks, it is limited to modeling outputs as distributions within the exponential family and is less general than PaiDEs. Furthermore, they only consider regression tasks with 1D outputs as their method is Bayesian and more computationally expensive.

# 6 CONCLUSION

To the best of our knowledge, we are the first to address the problem of epistemic uncertainty estimation for conditional diffusion models. Large generative models are becoming increasingly prevalent, and thus insight into the generative process is invaluable. We achieve this by introducing the DECU framework, which leverages an efficient ensembling technique and Pairwise-Distance Estimators (PaiDEs) to estimate epistemic uncertainty efficiently and effectively. Our experimental results on the ImageNet dataset showcase the effectiveness of DECU in estimating epistemic uncertainty. We explore per-pixel uncertainty in generated images, providing a fine-grained analysis of epistemic uncertainty. As the field of deep learning continues to push the boundaries of generative modeling, our framework provides a valuable tool for enhancing the interpretability and trustworthiness of large-scale generative models.

## Acknowledgements

The research leading to these results has received funding from the Horizon Europe Programme under the SAFEXPLAIN Project (https://www.safexplain.eu), grant agreement num. 101069595 and the Horizon Europe Programme under the AI4DEBUNK Project (https://www.ai4debunk.eu), grant agreement num. 101135757. Additionally, this work has been partially supported by PID2019-107255GB-C21 funded by MCIN/AEI/10.13039/501100011033 and JDC2022-050313-I funded by MCIN/AEI/10.13039/501100011033al by European Union NextGenerationEU/PRTR.

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

# Shedding Light on Large Generative Networks:
# Estimating Epistemic Uncertainty in Diffusion Models
# (Supplementary Material)

**Lucas Berry**[1,2]                **Axel Brando**[3]                **David Meger**[1,2]

[1]School of Computer Science, McGill University, Montreal, Quebec, Canada
[2]Centre for Intelligent Machines, McGill University, Montreal, Quebec, Canada
[3]Barcelona Supercomputing Center - Centro Nacional de Supercomputación (BSC-CNS), Spain

## A    COMPUTE AND HYPERPARAMETER DETAILS

We employed the same set of hyperparameters as detailed in Rombach et al. [2022] while training our ensemble of diffusion models. To facilitate this, we utilized their codebase available at (https://github.com/CompVis/latent-diffusion), making specific modifications to incorporate DECU. It's important to note that we specifically adopted the LDM-VQ-8 version of latent diffusion, along with the corresponding autoencoder, which maps images from 256x256x3 to 64x64x3 resolution. Our training infrastructure included an AMD Milan 7413 CPU clocked at 2.65 GHz, boasting a 128M cache L3, and an NVidia A100 GPU equipped with 40 GB of memory. Each ensemble component was trained in parallel and required 7 days of training with the specified computational resources. Our code is available at the following link.

## B    DATA

In the *binned classes* dataset, classes were randomly selected for each bin, and the images for each component were also chosen at random from the respective classes. In contrast, the *masked classes* dataset employed a clustering approach that grouped class labels sharing the same hypernym in WordNet. This grouping strategy aimed to bring together image classes with similar structures; for instance, all the dog-related classes were clustered together. Subsequently, each ensemble component randomly selected hypernym clusters until each component had a minimum of 595 classes. Note that each class was seen by at least two components.

## C    IMAGE GENERATION AND BRANCH POINT

In addition to the summary statistics concerning image diversity based on the branching point, we also provide visualizations of these effects in Figure 12, and Figure 13. These illustrations highlight the observation that bins with higher values tend to produce more consistent images that closely match their class label across all branching points. This distinction is particularly noticeable when comparing bin 1300 to bin 1. Furthermore, as the branching point increases, a greater variety of images is generated across all bins.

## D    LIMITATIONS

DECU has potential for generalization to other large generative models. However, it's important to note that applying PaiDEs for uncertainty estimation requires the conditional distribution of the output to be probability distribution with a known pairwise-distance formula. This requirement is not unusual, as some generative models, such as normalizing flows, produce known distributions as their base distribution [Tabak and Vanden-Eijnden, 2010, Tabak and Turner, 2013, Rezende and Mohamed, 2015].

Furthermore, our ensemble-building approach is tailored to the latent diffusion pipeline but can serve as a logical framework for constructing ensembles in the conditional part of various generative models. There's also potential for leveraging

low-rank adaption (LoRA) to create ensembles in a more computationally efficient manner [Hu et al., 2021]. However, it's worth mentioning that using LoRA for ensemble construction raises open research questions, as LoRA was originally developed for different purposes and not specifically designed for ensemble creation.

# E   UNCERTAINTY & BRANCH POINT

Assuming that the distributional distances between ensemble components grow as one progresses through the reverse process, similar to other models with similar dynamics [Chua et al., 2018], we can demonstrate the following: if $\lim_{D(p_i||p_j) \to \infty}$ for $i \neq j$, then $\hat{I}_\rho(y, \theta|x) = -\ln \frac{1}{M}$.

*Proof.*

$$\hat{I}_\rho(y, \theta|x) = -\sum_{i=1}^{M} \pi_i \ln \left[ \sum_{j=1}^{M} \pi_j \exp(-D(p_i||p_j)) \right]$$

$$= -\sum_{i=1}^{M} \pi_i \ln \left[ \pi_j \exp(-D(p_i||p_i)) + \sum_{j \neq i} \pi_j \exp(-D(p_i||p_j)) \right]$$

$$= -\sum_{i=1}^{M} \pi_i \ln \left[ \pi_j \exp(0) + \sum_{j \neq i} 0\pi_j \right]$$

$$= -\sum_{i=1}^{M} \pi_i \ln \left[ \frac{1}{M} \exp(0) \right]$$

$$= -\sum_{i=1}^{M} \frac{1}{M} \ln \left[ \frac{1}{M} \right]$$

$$= -\ln \left[ \frac{1}{M} \right]$$

□

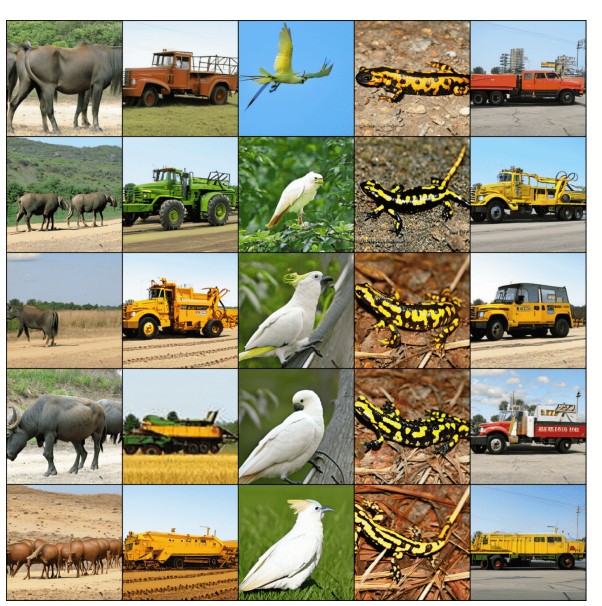 VS 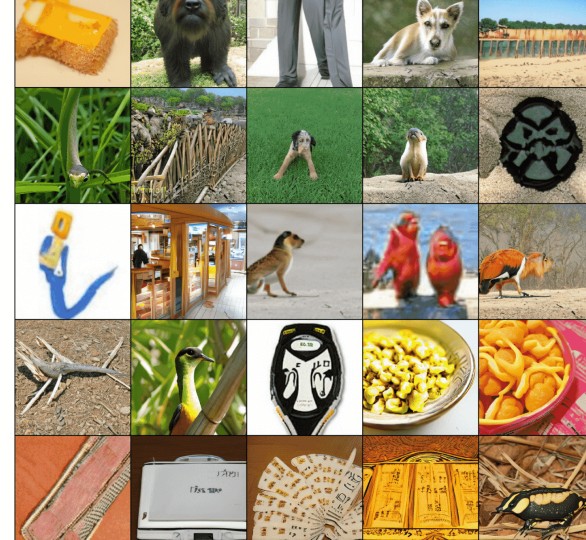

Figure 9: The left image showcases an example of image generation for five class labels with low epistemic uncertainty (bin 1300), arranged from left to right: water buffalo, harvester, sulphur crested cockatoo, european fire salamander, tow truck. The right image illustrates an example of image generation for five class labels with high epistemic uncertainty (bin 1), arranged from left to right: pedestal, slide rule, modem, space heater, gong. Note that each row corresponds to an ensemble component and $b = 1000$.

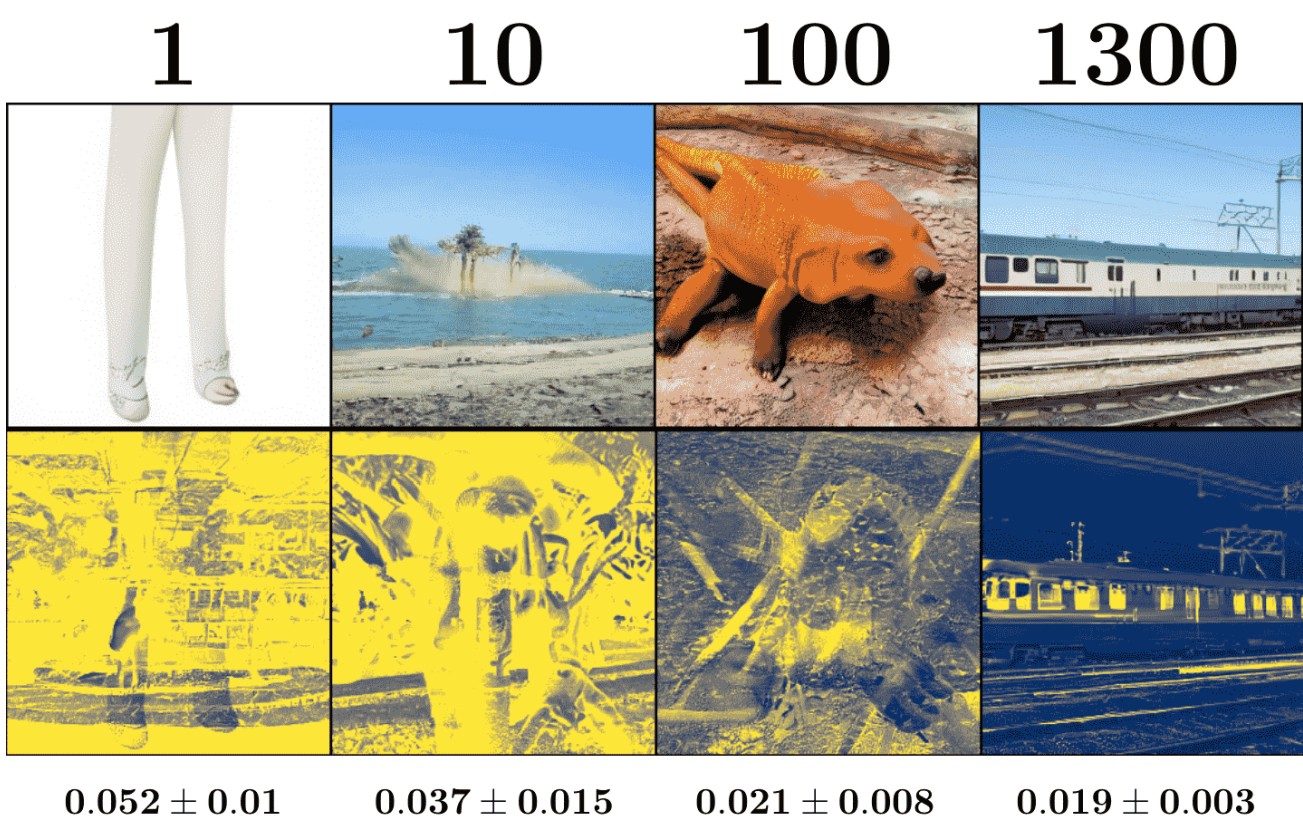

Figure 10: This shows the pixel uncertainty (high uncertainty in yellow and low uncertainty in blue) for one category from each bin, from left to right: cocktail shaker, howler monkey, Dungeness crab, bullet train. The number below the images shows the mean estimated $I(z_0, \theta | z_5, x, b = 5) \pm$ one standard deviation.

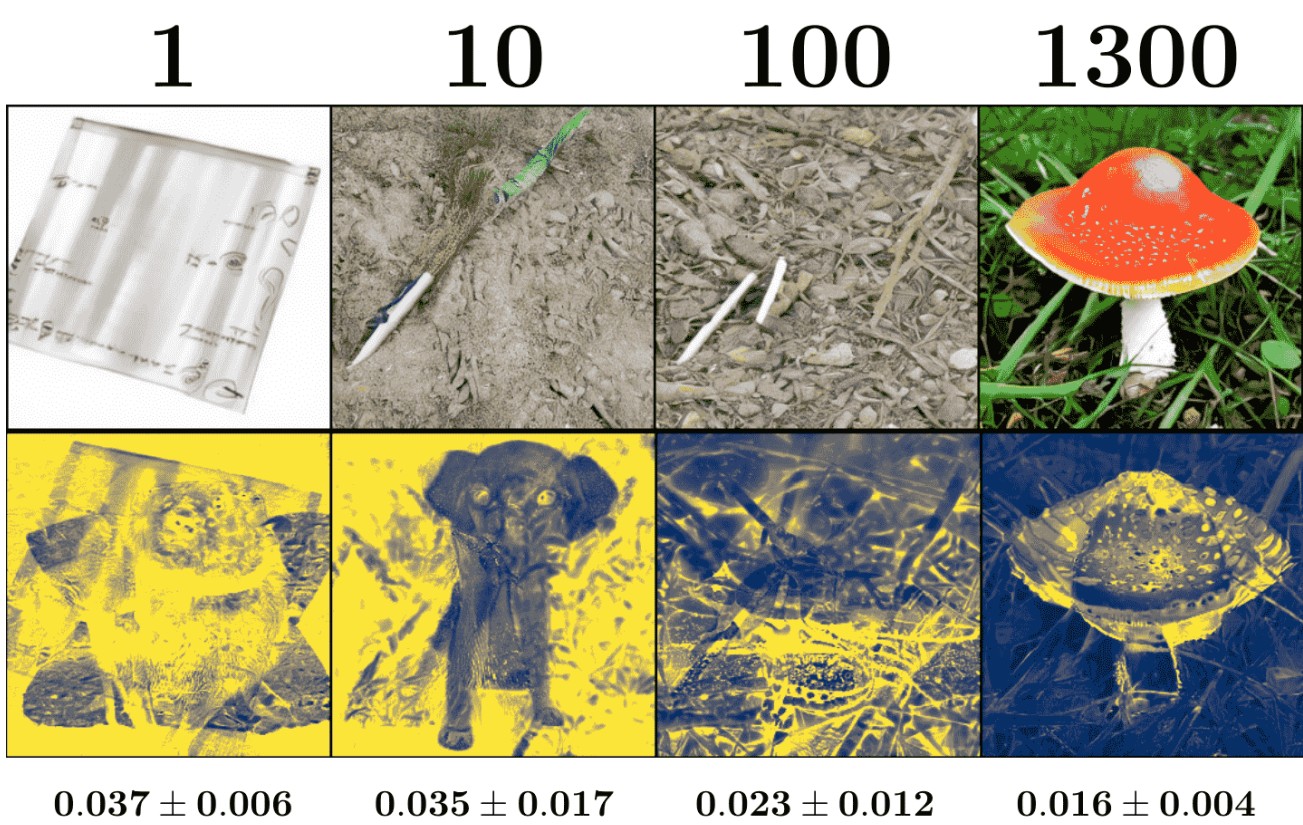

Figure 11: This shows the pixel uncertainty (high uncertainty in yellow and low uncertainty in blue) for one category from each bin, from left to right: grey whale, knot, terrapin, agaric. The number below the images shows the mean estimated $I(z_0, \theta | z_5, x, b = 5) \pm$ one standard deviation.

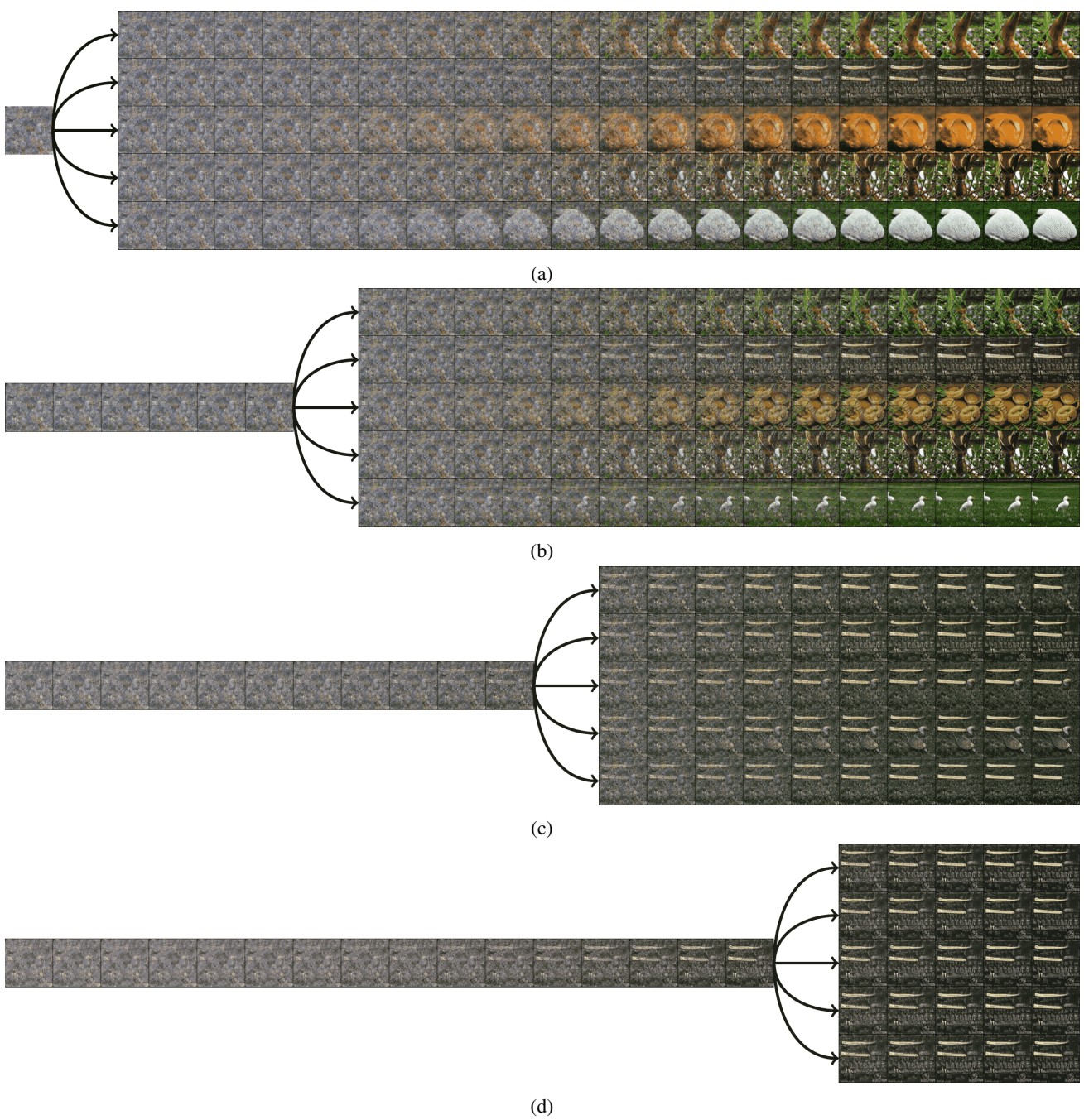

Figure 12: Image generation progression through the diffusion model for the class label marmoset from bin 100 for each branching point: (a) 1000, (b) 750, (c) 500, (d) 250.

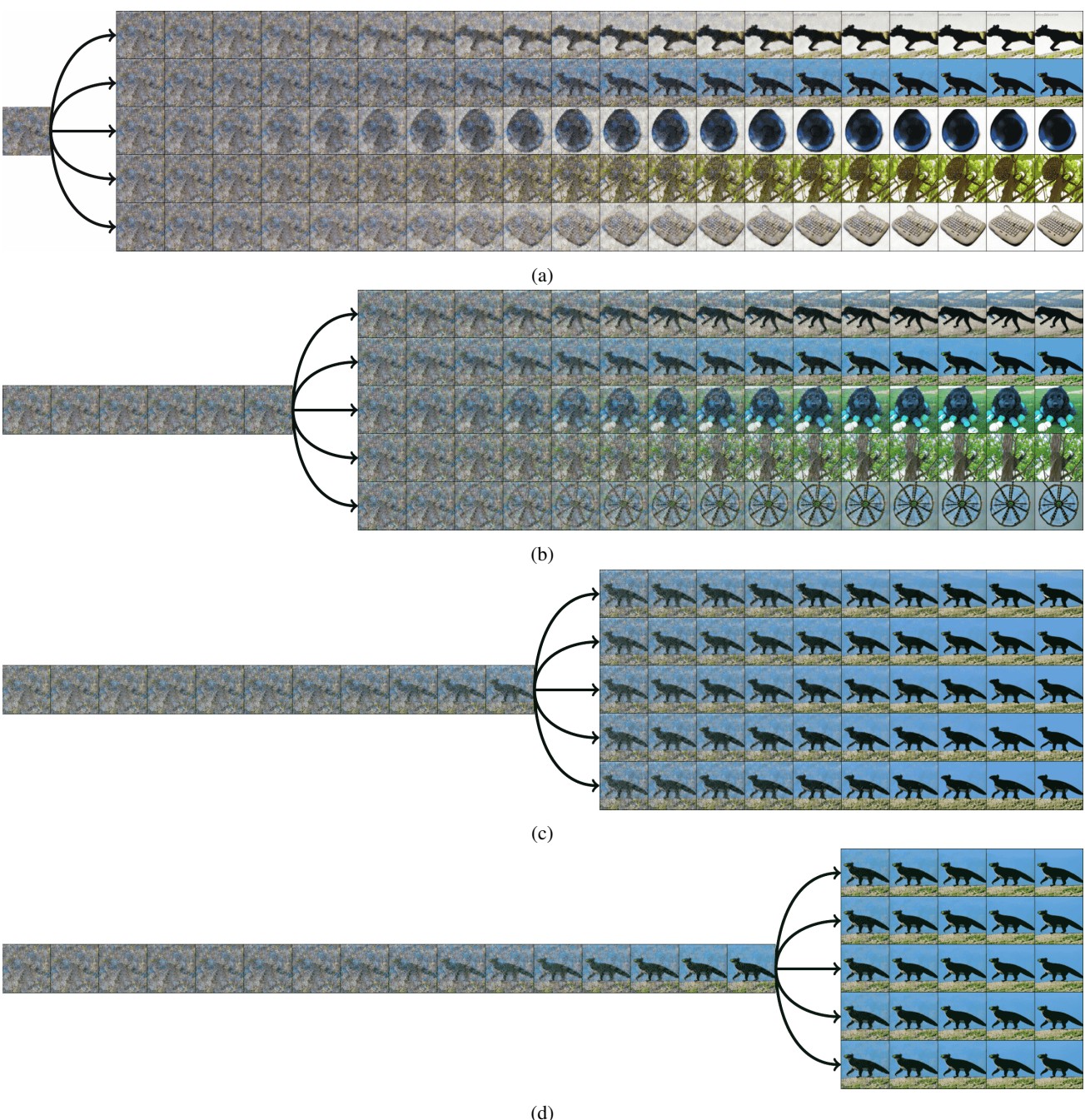

Figure 13: Image generation progression through the diffusion model for the class label steel arch bridge from bin 10 for each branching point: (a) 1000, (b) 750, (c) 500, (d) 250.