# OpenReview forum: "Shedding Light on Large Generative Networks: Estimating Epistemic Uncertainty in Diffusion Models"
_auai.org/UAI/2024/Conference — UAI 2024 poster_

### Official Review · Reviewer_YfPv · 2024-03-02

**Q2-1 Originality-Novelty:** 3
**Q2-2 Correctness-Technical Quality:** 3
**Q2-5 Clarity Of Writing:** 2

**Q10 Ethical Concerns:**

no ethical concerns.

**Q1 Summary And Contributions:**

This paper proposes a novel epistemic uncertainty estimation method for the diffusion model in high dimensional space (i.e., 256x256 resolution space). To the best of our knowledge, this is the first paper to explore calculating the epistemic uncertainty under the large diffusion model in such high dimensional space, especially using the ensemble based uncertainty estimation methods since it aggravates the computation cost and estimating the epistemic uncertainty in high dimensional space requires high computation cost. To solve it, the author first proposes an ensemble framework such that we just need to train a small part of the components and avoid training various UNets. Then, this paper further proposes the PaiDEs estimation based method to calculate the epistemic uncertainty.

**Q2-3 Extent To Which Claims Are Supported By Evidence:**

2: Fair: the main claims are somewhat supported by evidence (but the experimental evaluation may be weak, or does not match entirely with the claims, important baselines may be missing, proofs contain important ideas but lack rigor, algorithmic details are only discussed superficially, references are imprecise, assumptions are not sufficiently motivated or explicated, etc.).

**Q2-4 Reproducibility:**

3: Good: key resources (e.g. proofs, code, data) are available and key details (e.g. proofs, experimental setup) are sufficiently well-described for competent researchers to confidently reproduce the main results.

**Q3 Main Strengths:**

1. This paper focuses on an important problem for the diffusion model. Estimating epistemic uncertainty could avoid the diffusion model concentrating on the special data distribution and improve the generation of the diffusion model during the training stage.

2. The overall design of the proposed DECU is sound. Only training the components with random initialization to construct the ensemble methods is motivated since it reduces the computation cost.

3. The experimental results show that the proposed method could correctly estimate the epistemic uncertainty.

**Q4 Main Weakness:**

1. There is less detail about the trainable components in the Methodology part including Figure 2. The main contribution of this paper is estimating the uncertainty estimation via low computation cost and the trainable components in this paper are key for entire methods since the overall methods highly depend on the ensemble strategy and directly influence the final results.

2. It may be better to offer a simple theoretical proof of why epistemic uncertainty will be convergent into $-\log \frac{1}{M}$, where $M$ is the number of models in ensemble strategy. The experimental results shown in Table 1 show that SSIM has a different trend with different values of $b$. If $b$ could be convergent into $-\log \frac{1}{M}$, at least, SSIM will have less gap with different $b$.

3. There are some writing mistakes such as some typos. It may be better to keep the consistency for some symbols. Specifically, in Figure 5, $I_{W}(z_{0},\theta|z_{5},x,b=5)$ is confused although we think that the author wants to represent DDIM will scale the timesteps from $T=1000$ to $T=200$ and the time interval is 5. Directly using $I_{W}(z_{0},\theta|z_{1},x,b=5)$ is enough.

**Q5 Detailed Comments To The Authors:**

The main concern is Weakness. 1-2. Meanwhile, improving PaiDE may not be a major contribution to this paper. Thus, replacing it with more details about the proposed trainable components may be better. The current statement will limit the scope of the proposed method since the assumption of zero variance matrix $\Sigma$ is only for the DDIM sampler, which will be broken by other advanced samplers such as DPM-solver++. This question will not influence the score and is just a discussion.

**Q9 Complying With Reviewing Instructions:**

Yes

---

> ### Author Rebuttal · Authors · 2024-04-03
>
> Thank you for your careful review of our manuscript.
>
> Weakness 1
>
> We utilize a pre-trained latent diffusion model, consisting of over 456M parameters, wherein all weights in the autoencoder and UNet are kept static, and only the conditional embedding layer is trained. For each component, the encoder, decoder and UNets are shared, while the conditional embedding layer is randomly initialized and trained on different subsets of the training data during training. Consequently, the ensemble is formed within the embedding layer, which comprises 512K parameters and takes the class label as input. To further clarify this process, we have added the following text to the caption of Figure 2.
>
> >Our ensembles are constructed within the embedding layer, which accepts the class label as input. We create diversity through random initialization and by training each component on different subsets of the data. The encoders, decoders, and UNets for each component are shared, and we leverage pretrained networks from Rombach et al. [2022]. Notably, this reduces the number of parameters requiring training from 456 million to 512 thousand.
>
> To visually emphasize this, we have added dotted lines around Component 1 and Component 2 in Figure 2.
>
> Weakness 2
>
> Please note that in Figure 3, the branching point $b$ is fixed at 1000 and for one bin (1300), whereas the results presented in Table 1 indicate the similarity of images for different branching points across various bins. These two aspects are not necessarily comparable. We acknowledge the necessity for more theoretical grounding in our results, and thus, we have included the following addition to the text:
>
> >Assuming that the distributional distances between ensemble components grow as one progresses through the reverse process, similar to other models with similar dynamics [Chua et al., 2018], we can demonstrate the following: if $\lim_{D(p_i||p_j) \to \infty}$ for $i \neq j$, then $\hat{I}_{\rho}(y, \theta|x) = -\ln\frac{1}{M}$.
> >
> >$\\hat{I}_{\\rho}(y, \\theta|x)$
> >
> >$=-\sum_{i=1}^M \pi_i\ln\left[\sum_{j=1}^M \pi_j\exp(-D(p_i||p_j))\right]$
> >
> >$=-\sum_{i=1}^M \pi_i\ln\left[\pi_j\exp(-D(p_i||p_i))+\sum_{j\neq i} \pi_j\exp(-D(p_i||p_j))\right]$
> >
> >$=-\sum_{i=1}^M \pi_i\ln\left[\pi_j\exp(0)+\sum_{j\neq i} 0\pi_j\right]$
> >
> >$=-\sum_{i=1}^M \pi_i\ln\left[\frac{1}{M}\exp(0)\right]$
> >
> >$=-\sum_{i=1}^M \frac{1}{M}\ln\left[\frac{1}{M}\right]$
> >
> >$=-\ln\left[\frac{1}{M}\right]$
>
>
> Weakness 3
>
> Thank you for pointing this out, we have adopted your suggestion in order to clarify the text.

---

### Official Review · Reviewer_eUgg · 2024-03-12

**Q2-1 Originality-Novelty:** 3
**Q2-2 Correctness-Technical Quality:** 3
**Q2-5 Clarity Of Writing:** 3

**Q1 Summary And Contributions:**

In this paper, the authors introduce DECU for estimating epistemic uncertainty for diffusion models that efficiently trains ensembles of conditional diffusion models by incorporating a static set of pretrained parameters reducing the computational burden and the number of parameters. DECU employs PaiDEs to accurately measure epistemic uncertainty. The method was tested on ImageNet dataset.

**Q2-3 Extent To Which Claims Are Supported By Evidence:**

2: Fair: the main claims are somewhat supported by evidence (but the experimental evaluation may be weak, or does not match entirely with the claims, important baselines may be missing, proofs contain important ideas but lack rigor, algorithmic details are only discussed superficially, references are imprecise, assumptions are not sufficiently motivated or explicated, etc.).

**Q2-4 Reproducibility:**

2: Fair: key resources (e.g. proofs, code, data) are unavailable but key details (e.g. proof sketches, experimental setup) are sufficiently well-described for an expert to confidently reproduce the main results.

**Q3 Main Strengths:**

The authors establish the framework of DECU for class conditioned diffusion models and provide an evaluation of image diversity within DECU.
The paper was well organized. Overall the paper is sound and the results are presented in an appropriate way. Figures are illustrative and tables are comprehensive. The method was tested on various datasets.

**Q4 Main Weakness:**

Is it possible to test the model on other datasets and do any comparison with other approaches? I know it is the first work to address this uncertainty problem. Is there any ablation study?

**Q5 Detailed Comments To The Authors:**

same as weakness

**Q9 Complying With Reviewing Instructions:**

Yes

---

> ### Author Rebuttal · Authors · 2024-04-03
>
> Thank you for your careful review of our manuscript.
>
> ### Reviewer:
>
> “Is it possible to test the model on other datasets and do any comparison with other approaches?”
>
> ### Author’s Response:
>
> While we appreciate the suggestion to test our model on other datasets and compare it with alternative approaches, we'd like to highlight that our work represents the initial attempt to tackle this particular problem. Thus, at this early stage, the landscape of comparable methodologies is not clear. However, we firmly believe that in the context of the increasing utilization of Diffusion Models across various applications, the estimation of epistemic uncertainty holds paramount importance.
>
> In our endeavor to address this challenge, we deliberately selected the ImageNet dataset due to its complexity and profound relevance within the field. By focusing on such a widely recognized benchmark, we aimed to establish a solid foundation for our methodology's evaluation and validation.
>
> ### Reviewer:
>
> “Is there any ablation study?”
>
> ### Author’s Response:
>
> The computational demands of diffusion models made such studies infeasible within our current constraints. Specifically, our models necessitated a substantial training period of 7 days, not including debugging time, due to their resource-intensive nature. As a result, while we weren't able to conduct ablation studies in this iteration, we focused our efforts on ensuring the robustness and effectiveness of the trained models.

---

### Official Review · Reviewer_dGyA · 2024-03-24

**Q2-1 Originality-Novelty:** 3
**Q2-2 Correctness-Technical Quality:** 3
**Q2-5 Clarity Of Writing:** 2

**Q1 Summary And Contributions:**

This paper proposes a novel method for efficiently training an ensemble of diffusion models that allow for pixel-wise uncertainty quantification at inference time.

**Q2-3 Extent To Which Claims Are Supported By Evidence:**

3: Good: the main claims are supported by convincing evidence (in the form of adequate experimental evaluation, proofs, (pseudo-)code, references, assumptions).

**Q2-4 Reproducibility:**

2: Fair: key resources (e.g. proofs, code, data) are unavailable but key details (e.g. proof sketches, experimental setup) are sufficiently well-described for an expert to confidently reproduce the main results.

**Q3 Main Strengths:**

1. Novel Framework for ensemble-based uncertainty quantification with diffusion models
2. The framework is computationally efficient
3. The experiments, results, and overall problem seem valuable to the community and are well-motivated in the paper.

**Q4 Main Weakness:**

Updated: upon discussion with the authors they have  clarified their motivation which is indeed sound.

**Q5 Detailed Comments To The Authors:**

1. You say that the variance for the backwards process transition density in [1] is set to 0 ( you say $\Sigma_\theta\left(z_t, t, x\right)=0$  is intentionally set to a zero matrix, irrespective of its inputs, aligning with the approach in [1] ...) I went through [1] and could not see the evidence. Could you be more precise? I was also unable to find this in [2].
2. Following up on the above, if you read [1] eq 62, you will see that $\Sigma_\theta\left(z_t, t, x\right)=\sigma_t$ as in vanilla DDPM for training and not $0$ looking at your equation 1 setting  $\Sigma_\theta\left(z_t, t, x\right)=0$ would render the DDPM objective singular so. Unfortunately, it sounds like this is not true and a huge misconception on the author's part.

Update: Variance is set to 0 at test time akin to prov flow ODE in score SDE models , thus not affecting training time.

Authors have clarified this confusion from the reviewers side. Personally ddim is a bit sloppy in the way this is handled as it’s possible to arrive at the DDIM integrator from the prob flow ODE + exponential integrator , rather than setting the volatility to 0 (see prop 2 here https://arxiv.org/pdf/2204.13902.pdf ) .

[1] Jiaming Song, Chenlin Meng, and Stefano Ermon. Denoising diffusion implicit models. In International Conference on Learning Representations, 2020.
[2] Rombach, R., Blattmann, A., Lorenz, D., Esser, P. and Ommer, B., 2022. High-resolution image synthesis with latent diffusion models. In Proceedings of the IEEE/CVF conference on computer vision and pattern recognition (pp. 10684-10695).

**Q9 Complying With Reviewing Instructions:**

Yes

---

> ### Author Rebuttal · Authors · 2024-04-02
>
> Thank you for your careful review of our manuscript.
>
> The reviewer notes that we have made conceptual errors with regards to the treatment of the covariance matrix in the reverse diffusion process. However, our treatment of the covariance matrix is based on previously published and widely accepted works in the field of diffusion models (see references 1-4). In [1], the authors state that the covariance matrix can equal 0 in section 4, paragraph 2:
>
> >We note another special case when $\sigma_t = 0$ for all $t$; the forward process becomes deterministic
> given $x_{t−1}$ and $x_0$, except for $t = 1$; in the generative process, the coefficient before the random
> noise $\epsilon_t$ becomes zero.
>
> This concept is reiterated in [3], section 2.1, paragraph 3.
>
> Notably, setting the covariance matrix to zero is a widely accepted practice known to enhance image quality, as evidenced by the implementation of [2]. Although the reviewer is correct that this is not explicitly stated in [2], the covariance matrix is set to zero in [2]'s code. In the foundational DDPM paper [4], the authors create an objective function (Equation 14), which we employ in the current paper. Importantly, this objective is independent of the parameter $\sigma_t$ and thus does not lead to singularity when $\sigma_t=0$.
>
> We trust that these references provide clarity on the matter and that subsequent points do not need to be addressed, as these were contingent on the conceptual flaw related to the covariance matrix.
>
> [1] Jiaming Song, Chenlin Meng, and Stefano Ermon. Denoising diffusion implicit models. In International Conference on Learning Representations, 2020.
>
> [2] Rombach, R., Blattmann, A., Lorenz, D., Esser, P. and Ommer, B., 2022. High-resolution image synthesis with latent diffusion models. In Proceedings of the IEEE/CVF conference on computer vision and pattern recognition (pp. 10684-10695).
>
> [3] Prafulla Dhariwal and Alex Nichol. Diffusion models beat gans on image synthesis. In NeurIPS, 2022.
>
> [4] Jonathan Ho, Ajay Jain, and Pieter Abbeel. Denoising diffusion probabilistic models.
> arXiv:2006.11239, 2020.

---

### Official Review · Reviewer_VEeg · 2024-03-24

**Q2-1 Originality-Novelty:** 2
**Q2-2 Correctness-Technical Quality:** 3
**Q2-5 Clarity Of Writing:** 3

**Q1 Summary And Contributions:**

- The paper introduces Diffusion Ensembles for Capturing Uncertainty (DECU), a novel framework for estimating epistemic uncertainty in large generative diffusion models, which are challenging for traditional methods due to their computational complexity and parameter count.
- DECU utilizes an efficient ensembling technique that leverages a static set of pretrained parameters, reducing the computational load and the number of parameters needing training, and employs Pairwise-Distance Estimators (PaiDEs) to measure epistemic uncertainty.
- The framework is demonstrated to be effective through experiments on the ImageNet dataset, particularly in capturing epistemic uncertainty in under-sampled image classes, and provides a fine-grained analysis of per-pixel uncertainty in generated images.
- DECU's approach is significant for applications where understanding and interpreting the outputs of generative models are critical, such as in medical imaging and self-driving systems, and it aligns with international safety standards for automated systems.
- The paper's contributions include establishing the DECU framework, assessing its effectiveness on the ImageNet dataset, and evaluating image diversity within DECU, thereby illuminating the area of epistemic uncertainty in conditional diffusion models.

**Q2-3 Extent To Which Claims Are Supported By Evidence:**

2: Fair: the main claims are somewhat supported by evidence (but the experimental evaluation may be weak, or does not match entirely with the claims, important baselines may be missing, proofs contain important ideas but lack rigor, algorithmic details are only discussed superficially, references are imprecise, assumptions are not sufficiently motivated or explicated, etc.).

**Q2-4 Reproducibility:**

2: Fair: key resources (e.g. proofs, code, data) are unavailable but key details (e.g. proof sketches, experimental setup) are sufficiently well-described for an expert to confidently reproduce the main results.

**Q3 Main Strengths:**

- Introduces DECU, a novel framework for estimating epistemic uncertainty in large generative diffusion models, addressing a gap in the understanding of these complex systems.
- DECU leverages an efficient ensembling technique that significantly reduces computational burden by incorporating a static set of pretrained parameters, making it more feasible to train on large datasets like ImageNet.
- Employs Pairwise-Distance Estimators (PaiDEs) that are capable of accurately measuring epistemic uncertainty in high-dimensional spaces, which is crucial for the analysis of under-sampled image classes.
- Provides a fine-grained analysis of epistemic uncertainty at the per-pixel level in generated images, enhancing the interpretability and trustworthiness of generative models.
- Demonstrates the effectiveness of the DECU framework through experiments on the ImageNet dataset, showcasing its capability to capture epistemic uncertainty, particularly in classes with fewer samples.

**Q4 Main Weakness:**

- The DECU framework may not scale efficiently to datasets larger than ImageNet due to the computational demands of training ensembles of diffusion models, even with a static set of pretrained parameters.
- While DECU employs Pairwise-Distance Estimators (PaiDEs) for uncertainty estimation, the choice of distance metrics and their sensitivity to high-dimensional spaces could affect the accuracy of uncertainty quantification.
- The paper does not discuss the potential for overfitting to the ImageNet dataset, which could limit the generalizability of the DECU method to other datasets or real-world scenarios.
- There is a lack of comparison with other uncertainty estimation methods, which would be necessary to fully understand the strengths and weaknesses of DECU relative to existing approaches.
- No source code of the implementation provided.

**Q5 Detailed Comments To The Authors:**

- Explore the use of alternative uncertainty estimation metrics beyond Pairwise-Distance Estimators to potentially enhance the robustness and accuracy of uncertainty quantification in high-dimensional spaces.
- Conduct comparative studies with other uncertainty estimation methods to benchmark DECU's performance and identify areas for refinement.
- Investigate the scalability of the DECU framework to larger and more diverse datasets to ensure its applicability across various domains.
- Address the potential for overfitting by incorporating regularization techniques or cross-validation strategies to improve the generalizability of the framework.
- Consider the integration of active learning strategies to further reduce the data requirements for training diffusion models, which could make the DECU framework more practical for real-world applications.

**Q9 Complying With Reviewing Instructions:**

Yes

---

> ### Author Rebuttal · Authors · 2024-04-03
>
> Thank you for your careful review of our manuscript. The suggestions concerning using additional datasets to validate DECU’s methodology are indeed intriguing. However, the potential challenges associated with scaling the DECU framework to datasets larger than ImageNet, particularly due to the significant computational demands—requiring up to 7 days to train our models—, render the implementation of this idea impractical. Moreover, the absence of baselines in this paper reflects the current lack of research on uncertainty estimation for diffusion models. Please note that source code is included with our submission and will be made public upon acceptance.

---

### Official Review · Reviewer_xggP · 2024-03-27

**Q2-1 Originality-Novelty:** 1
**Q2-2 Correctness-Technical Quality:** 3
**Q2-5 Clarity Of Writing:** 2

**Q1 Summary And Contributions:**

This paper proposes to translate Pairwise distance estimator (recently introduced by another paper into an estimate of a component of uncertainty) for informing diffusion model.

**Q2-3 Extent To Which Claims Are Supported By Evidence:**

2: Fair: the main claims are somewhat supported by evidence (but the experimental evaluation may be weak, or does not match entirely with the claims, important baselines may be missing, proofs contain important ideas but lack rigor, algorithmic details are only discussed superficially, references are imprecise, assumptions are not sufficiently motivated or explicated, etc.).

**Q2-4 Reproducibility:**

2: Fair: key resources (e.g. proofs, code, data) are unavailable but key details (e.g. proof sketches, experimental setup) are sufficiently well-described for an expert to confidently reproduce the main results.

**Q3 Main Strengths:**

Hot topic, it seems important to address uncertainty in conditional diffusion model

**Q4 Main Weakness:**

- Not novel per se (at least in the way the authors present)
- no comparison to any benchmark alternative, it's take it as it is.
- no theory, analysis, justification bound, you name it. Again take it as it is.

**Q5 Detailed Comments To The Authors:**

First, I'm low confident here not having worked myself on diffusion model, so take my review as the opinion of an outsider.
Unfortunately that opinion is not glowing: I can totally understand the motivation, but seeing an idea from another paper being importer to a problem and presented with one run that report numbers outside of any comparison just feel like below the bar of scientific progress. What was exactly the problem we address? And how do we know we did?

I get it that ML and GenAI are hard to validate, that theory is arduous and empirical evaluation expensive, I sympathize and I do think the authors should pursue the idea. At this point I just don't see why reporting on their experience to do it would inform somebody else. Note that not being an insider, I may apply acceptance condition that are not expected for work of that nature. But that's unfortunately the only thing I can do.

**Q9 Complying With Reviewing Instructions:**

Yes

---

### Meta-Review · Area_Chair_8mGB · 2024-04-19

This paper proposes DECU, a novel framework for estimating epistemic uncertainty in diffusion models, which are known to be computationally expensive to train for large datasets like ImageNet.

Strengths:

The paper identifies a crucial challenge in diffusion models: the lack of methods to quantify epistemic uncertainty. DECU represents a first attempt at tackling this issue.
The proposed framework, DECU, utilizes an ensemble approach with static pre-trained parameters, reducing computational burden compared to training individual diffusion models from scratch. It also employs Pairwise-Distance Estimators (PaiDEs) to measure epistemic uncertainty in high-dimensional spaces.
The authors demonstrate DECU's effectiveness on the ImageNet dataset, showcasing its ability to capture uncertainty, particularly in under-sampled classes. They also provide a per-pixel analysis of uncertainty in generated images.

Weaknesses:

While DECU is novel in its application to diffusion models, reviewers noted that the core ideas of ensembles and PaiDEs have been explored in prior work.
The paper lacks comparisons with existing uncertainty estimation methods, making it difficult to assess DECU's relative strengths and weaknesses.
Concerns were raised regarding the scalability of DECU to even larger datasets and its generalizability to scenarios beyond ImageNet. The authors acknowledge the computational demands but highlight the importance of uncertainty estimation for real-world applications.
The paper could benefit from a more rigorous theoretical foundation, including proofs or justifications for specific choices.
Reviewers identified areas where the paper could be improved in terms of clarity and detail, such as a more thorough description of the trainable components and a clearer explanation of the notation used.

Recommendation:

Overall, this paper presents a promising approach to estimating epistemic uncertainty in diffusion models. A majority of reviewers vote for acceptance. Only one reviewer with a lack of experience in the domain area recommended rejection.